# SKVQ: Sliding-window Key and Value Cache Quantization for Large Language Models

**Haojie Duanmu**[*]
Shanghai AI Laboratory
Shanghai Jiao Tong University

**Zhihang Yuan**[*]
Houmo AI

**Xiuhong Li**[†]
Peking University

**Jiangfei Duan**
CUHK
Shanghai AI Laboratory

**Xingcheng Zhang**
Shanghai AI Laboratory

**Dahua Lin**
CUHK
Shanghai AI Laboratory

## Abstract

Large language models (LLMs) have demonstrated the capability to process extended token sequences, enabling complex tasks such as book comprehension and long-form text generation. However, as context length increases, the key-value (KV) cache required for LLMs consumes substantial memory, becoming a bottleneck for deployment. This paper introduces SKVQ (Sliding-window KV cache Quantization), a strategy designed to address the challenge of extremely low bitwidth KV cache quantization. SKVQ re-arranges the channels of the KV cache to enhance channel similarity within quantization groups and applies clipped dynamic quantization at the group level. Furthermore, SKVQ maintains high precision for the most recent window tokens in the KV cache, preserving accuracy for a small yet critical portion of the cache. Our evaluation of LLMs demonstrates that SKVQ achieves high compression ratios while maintaining accuracy, outperforming previous quantization methods. SKVQ enables the quantization of the KV cache to 2-bit keys and 1.5-bit values with minimal accuracy loss. This advancement allows processing context lengths of up to 1M tokens on an 80GB GPU for a 7B parameter model, resulting in up to 7 times faster decoding.

## 1 Introduction

Large Language Models (LLMs) have recently demonstrated remarkable success in artificial intelligence. As LLMs advance, the demand for extended context support has increased. For instance, OpenAI GPT-4 Turbo can handle 128k tokens (Achiam et al., 2023), and Google Gemini 1.5 can process up to 1 million tokens (Team et al., 2023). This expanded token capacity enables LLMs to address more complex tasks, including book comprehension, large image analysis, and video processing, enhancing their versatility. LLM inference operates in an auto-regressive manner, generating sentences token by token. To reduce the computation overhead, inference system always store key and value activations in memory known as the key-value (KV) cache, for reuse during subsequent token generation. With the increasing popularity of utilizing LLM for long sequence tasks, the KV cache consumes a significant amount of memory. On the other hand, the large amount of KV cache can also bring a large amount of memory access in the attention mechanism when generating the output tokens. The system will be stuck on the memory access, known as the memory-bound problem in LLM inference (Yuan et al., 2024).

To tackle the problem of large KV cache size in language models, several compression techniques have been proposed. One approach is KV eviction (Zhang et al., 2023), which

---

[*]Equal Contribution
[†]Correspondence to: lixiuhong@pku.edu.cn

involves removing less important key-value pairs from the cache to free up space. However, this may impact the accuracy of inference. Another method is KV offloading (Sheng et al., 2023), which transfers a portion of the KV cache to slower but larger storage devices like main memory and even secondary storage. However, this may slow down the system due to the low bandwidth of these devices. Scientists have recently been studying the compression of KV cache using quantization. This involves converting floating point KV cache, which initially utilizes a large number of bits, into a format that uses fewer bits. Several novel approaches have been developed to accomplish this, including KVQuant (Hooper et al., 2024), WKVQuant (Yue et al., 2024), and KIVI (Liu et al., 2024). Previous quantization methods have been successful in reducing memory requirements and the number of memory accesses. However, they faced a challenge when using very low-bitwidth quantization because it led to a significant decrease in accuracy, as shown in Figure 1.

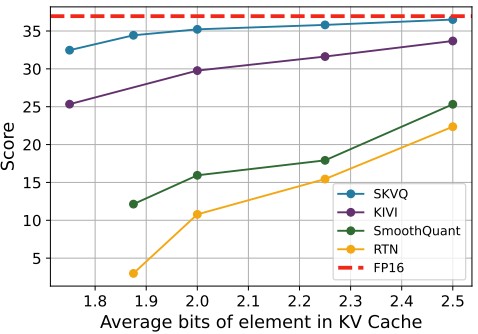

In this paper, we observe that there is a significant difference in the distribution of different channels during the quantization process. This has a great impact on quantization accuracy, especially in extremely low-bitwidth scenarios. To alleviate this problem, we propose the clipped dynamic quantization with channel reorder. First, we use a transformation invariant permutation to group similar channels based on their statistical characteristics. Second, we apply clipped dynamic quantization to further mitigate the outlier problem. In this way, we greatly reduce the quantization error within each group, thus improving the accuracy of the quantized model.

Figure 1: Results on GovReport and MultiFieldQA-zh (Mistral-7b-Instruct-V0.2). We count the storage for meta data including quantization params and reorder index.

Meanwhile, we discover that the protecting the accuracy of these small portion of but more important caches in KV cache quantization is critical. Due to the locality of attention, these recently generated KV caches are highly likely to be attended to with a high probability. We propose a sliding window quantization strategy. This mechanism preserves a small portion of the most recently generated KV cache from being quantized. After generating new tokens, the probability of attending to the old tokens' KV cache decreases significantly, so the accuracy loss caused by quantizing them is minimal. The proposed method is named as sliding-window KV cache quantization (SKVQ). It is efficient and easy to implement in existing inference system, which makes it practical for real-world deployment.

To evaluate the effectiveness of our method, we experiments on models of LLaMA (Touvron et al., 2023) and Mistral (Jiang et al., 2023) family. The experiments show that our methods can quantize the key cache into 2 bits and value cache into 1.5 bits with almost no accuracy drop. Compared with the previous quantization method, our approach can achieve optimal performance under different average bit widths as shown in Figure 1. Our performance analysis shows SKVQ enables 1M context length in a single A100-80GB for a 7b model. As for the inference latency, in the case of batch size 128 and sequence length 200k, the theoretical 7x speedup in decoding phase can be achieved [1].

## 2    Related Work

There are many multi-billion scale transformer quantization methods designed for LLMs. A main branch of LLM quantization is weight-only quantization, which only involves the quantization of model weights to lower precision. For instance, GPTQ (Frantar et al., 2022) uses second-order approximation to quantize weights, enabling the weight quantization of LLMs into 4-bit. AWQ (Lin et al., 2023) quantizes model weights to 4bits whith an activation-aware manner. SqueezeLLM (Kim et al., 2023) adopts the concept of sensitivity-based

---

[1]The performance analysis can be found in Appendix 9

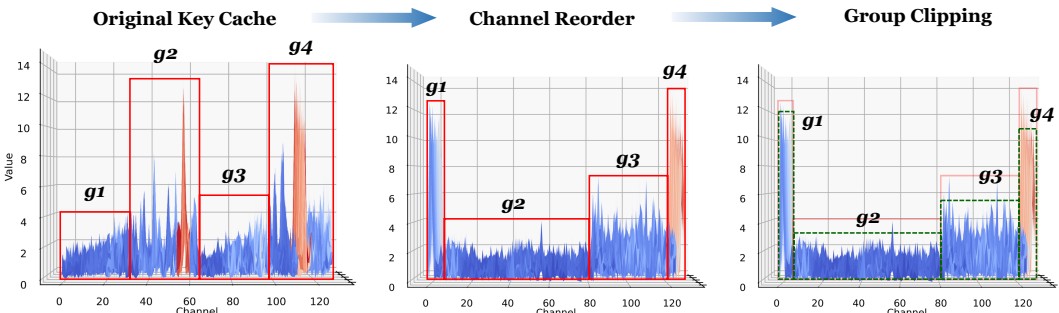

Figure 2: Visualization of the key cache going through channel reorder and group clipping in sequence. The elements in the red/green box will be placed in the same group to share the quantization parameters.

non-uniform quantization along with Dense-and-Sparse decomposition. This line of work is orthogonal to ours, as they can be combined together.

Another line of work focuses on weight-activation quantization. llm.int8() (Dettmers et al., 2022) retain outlier channel to full precision, so that other parts can be better compressed to 8bits. SmoothQuant (Xiao et al., 2022) uses equivalent transformations to balance the quantization complexity for both activation and weight, making the activation easier to quantize. RPTQ (Yuan et al., 2023) reorder the channels to reduce the variance in one quantization cluster, further enhancing the accuracy. ATOM (Zhao et al., 2023) improves quantization performance and reduces inference latency by using finer-grained quantization with an efficient kernel. However, since these works are not specifically designed for KV cache quantization, even applying the best results from such works still results in significant losses in KV cache compression. We compare with these works in the experimental section.

Recently, as natural language tasks require processing longer contexts, researchers have focused on quantizing key-value caches. Several new methods have been developed, such as KVQuant (Hooper et al., 2024), WKVQuant (Yue et al., 2024), and KIVI (Liu et al., 2024). Quantizing the KV cache can significantly reduce both the memory requirements and the number of memory accesses needed. Our experimental results show that the performance of our method on long context tasks performs the best in this type of work.

There are also a series of work dedicated to the design of KV cache eviction strategy (Liu et al., 2023; Ge et al., 2023; Zhang et al., 2023; Xiao et al., 2023). Unlike KV cache quantization, which retains all caches but compresses them to low precision, these methods selectively retain part of the KV cache and discard other caches directly. These methods usually allocate a fixed-size buffer for KV cache . When the generated KV cache exceeds the buffer limit, some tokens considered less important will be evicted from the buffer. These methods are inevitably and irrecoverably discarding KV pairs deemed, in one way or another, less important than others. Our approach is inspired by and can be well integrated with such work.

## 3 Method

### 3.1 Clipped Dynamic Quantization with Channel Reorder

Quantization is to transform the high-bitwidth float values into low-bitwidth integer values. The quantization process can be formulated as $\text{clamp}(\lfloor \frac{\mathbf{X}-z}{h} \rceil, 0, 2^N - 1)$, where $\mathbf{X}$ is the float values and $h$ is scaling factor and $z$ is zero point. Previous studies have highlighted significant variations in numerical values among activation channels (Xiao et al., 2022; Wei et al., 2022; 2023). As shown in Figure 2, we also observe substantial variations between channels and tokens in the KV cache (high channel variance). Therefore, directly quantizing the KV cache leads to substantial quantization errors. If values in different channels share the scaling factor and zero point, the value from outlier channels skew the quantization range. Especially in the low-bitwidth case, this makes almost all elements except outlier channel

quantize to the same value, and this loss of information leads to significant performance drop. To tackle this issue, we introduce channel transformation based quantization.

To address this problem, some methods have proposed using additional quantization parameters or keeping certain channels in float format to handle outliers (Dettmers et al., 2022). However, we have noticed that the concept of outliers is relative. The channels with the highest values are outliers compared to the medium-sized channels, and the medium-sized channels are outliers compared to the small-sized channels. Other methods propose smoothing the difference between channels by multiplying an extra factor before quantization (Shao et al., 2023; Yue et al., 2024). However, these methods do not take into account the differences in token dimension(sequence length dimension) of KV cache. The magnitude of values can vary between different tokens. We have observed that the variation in magnitude of non-outlier channels is relatively high. Specifically, some channels experience magnitude changes of several times or even dozens of times. Smoothing is not effective in addressing this phenomenon, especially in extremely low bitwidth quantization.

**Channel Reorder.** Inspired by RPTQ(Yuan et al., 2023), we employ a permutation invariant transformation and then apply group clipping to solve the problem of extremely low bitwidth quantization for KV cache . The permutation invariant transformation allows us to change the order of computation without changing a operation's output. For example, when we execute the matrix multiplication $S = Q \times K^T$, we can rearrange the columns of $Q$ and the rows of $K$ (which represent their channel dimension) in the same order without affecting the result of the computation.

We perform channel reorder on KV cache to make channels with similar data distribution are grouped together for quantization. Values in channels with similar distribution are quantized together. By this way, we can greatly reduce the quantization error of channels with smaller ranges. The same as Yuan et al. (2023), we do the corresponding equivalent permutation for $Q$ and $W_o$ to avoid explicit reorder operation. The calculation of attention module $O = \text{Softmax}(QK^T) \cdot V \cdot W_o$ is transformed as:

$$O = \text{Softmax}(P_k Q \cdot (K^T P_k^T)) \cdot P_v V \cdot W_o P_v^T \tag{1}$$

where $P_k \in \mathbb{R}^{C_{in} \times C_{in}}$ and $P_v \in \mathbb{R}^{C_{in} \times C_{in}}$ are channel reorder matrix of Key and Value respectively [2]. In our algorithm, the index is calculated based on the statistical characteristics of each channel. Specifically, we extract the distribution feature of each channel and then use the KMeans algorithm to cluster channels with similar characteristics into the same group. We also compared channel reordering with mathematical equivalent smoothing, and the results in Appendix 10 demonstrated the effectiveness of the former.

**Clipped Dynamic Quantization.** Dynamic per-token quantization is widely used method for quantizing the activations in LLMs (Xiao et al., 2022). Different with static quantization that use the static $h$ and $z$, dynamic quantization will compute new $h$ and $z$ using the maximum value and minimum value for each token: $h = \frac{\max(\mathbf{X}) - \min(\mathbf{X})}{2^N - 1}, z = \frac{\min(\mathbf{X})}{h}$.

Previous work about weight quantization (Lin et al., 2023; Shao et al., 2023) has shown that introducing clipping when quantizing weights can improve the quantization performance. According to the second picture in Figure 2, even though we have grouped similar channels together, there are inevitably some outliers within a quantization group. In order to reduce the impact of these outliers on other values in the same group, we propose the clipped dynamic quantization, which can be formulated as:

$$f(\alpha, \mathbf{X}) = \text{clamp}(\lfloor \frac{\mathbf{X} - z}{h} \rceil, 0, 2^N - 1), \text{where } h = \frac{\alpha(\max(\mathbf{X}) - \min(\mathbf{X}))}{2^N - 1}, z = \frac{\alpha \min(\mathbf{X})}{h}. \tag{2}$$

We introduce a clipping scale $\alpha \in (0, 1]$ for each group to compute $h$ and $z$. In order to get the best clipping scale, for each transformer block we try to minimize the MSE of the output of the attention module before and after quantization, i.e., the optimization objective:

$$\alpha^* = \arg\min_{\alpha} \mathcal{L}(\alpha), \quad \mathcal{L}(\alpha) = \text{MSE}(O^q, O) \tag{3}$$

---

[2]We fuse the channel reorder index into the the projection weight matrix of attention module. We describe the fusion in the Appendix 6.

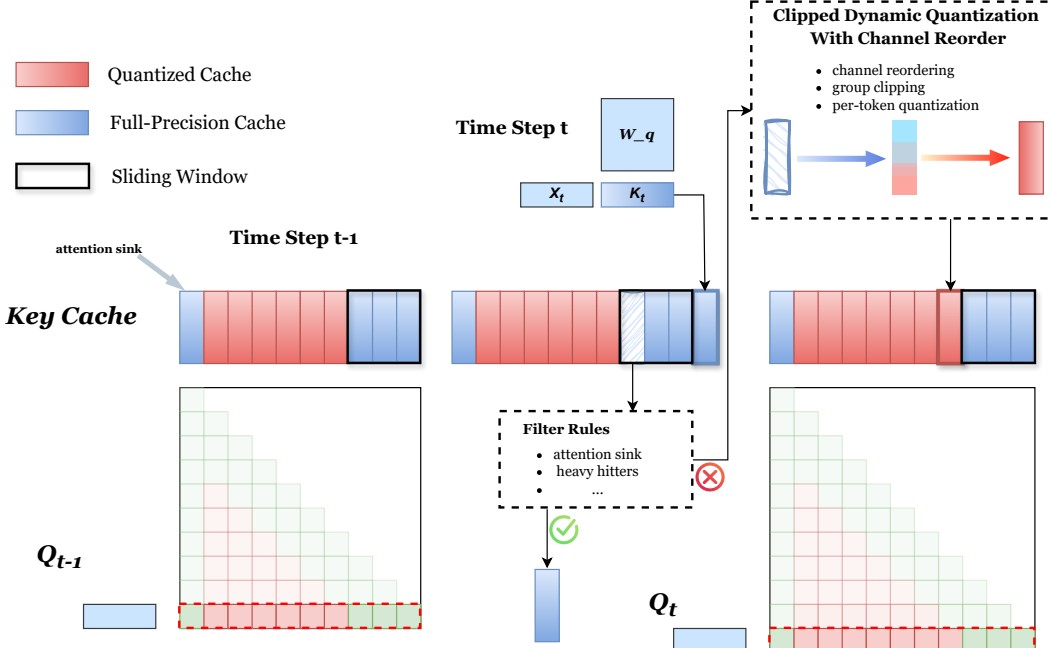

Figure 3: Overview of sliding window quantization Strategy. In each time step, we ensure the latest $w$ KV cache is full precision. For a token cache that slides out of the window, we make a decision based on the filter rules and choose whether to retain it to high precision.

where $O^q$ is the output of attention module after quantizing KV cache. Unlike weight-only quantization, the KV cache is generated at runtime, it is costly to solve this optimization for each inference. Therefore, we approximate it by offline calibration. By performing optimization on a calibration dataset in advance, we get the approximate $\hat{\alpha}^*$ for each group. We share the same $\hat{\alpha}^*$ across different tokens. Using the approximate $\hat{\alpha}^*$, we can also improve the quantization performance without introducing significant inference cost.

Using channel reorder and clipped dynamic quantization, the elements falling within the same group can more fully utilize the numerical range of the quantized data type, thus reducing the quantization error. Because all the parameter $P_k$, $P_v$, $\alpha$ is determined offline and the reorder operation can be fused into linear layers, it is efficient to implement the clipped dynamic quantization with channel reorder on existing inference frameworks.

## 3.2 Sliding Window Quantization Strategy

Although clipped dynamic quantization with channel reorder can improve the quantization performance to a large extent, extremely low-bitwidth KV cache quantization still suffers serious performance degradation, especially when the sequence length become longer. This is because the quantization errors accumulate along the sequence dimension. Because the auto-regressive manner of the LLM, the decoding of a new token depends on the previous KV generated. We realize that this not only a challenge but also a chance, the auto-regressive manner can be fully exploited to develop more flexible quantization strategies.

**Locality**. Previous research has demonstrated that attention modules exhibit strong locality (Kovaleva et al., 2019; Beltagy et al., 2020; Ge et al., 2023). This locality implies that at each time step, the attention module focuses more on recently generated tokens. We posit that in KV cache quantization, *preserving the accuracy of a small but critical portion of the cache is more important than maintaining the larger, less significant content from earlier in the sequence.*

Motivated by this observation, we propose a sliding window quantization strategy that maintains high precision for the most recent KV cache of a window of $w$ tokens. The workflow, illustrated in Figure 3, consists of two phases: 1) Prefill phase: For each transformer block, after generating the KV cache, we first compute attention using full-precision KV

cache. We then quantize the KV cache, reserving the last $w$ token cache pairs at full precision. 2) Decode phase: We process only the token that slides out of the window at each time step. This approach ensures lossless KV cache generation for each transformer block during the prefill phase. It also enhances the quality of generated content by leveraging the locality of the attention module in the decode phase.

**Important KV Cache Filter**. Beyond recently generated tokens, certain tokens are particularly sensitive to quantization. We explored additional methods to identify critical tokens whose KV cache should be maintained at high precision. Inspired by (Xiao et al., 2023), we recognized that the initial tokens of a prompt are crucial for the entire generation process. Consequently, we incorporated an attention sink into our filter rules, reserving the first few tokens at high precision. We observed that maintaining a small number of sink tokens at high precision is effective. Given the fixed positions of sink tokens, this approach is easily implementable and was enabled in our experiments. Some cache eviction methods, such as those proposed by (Liu et al., 2023; Zhang et al., 2023), monitor each token's cumulative attention score, treating these scores as token frequency and retaining only the most frequent tokens (heavy hitters) in the KV cache. While keeping heavy hitters at high precision seems intuitive, we did not implement this approach in our experiments for two reasons: 1) The improvement on prediction accuracy by keeping heavy hitters high precision is not significant. 2) When using FlashAttention (Dao, 2023), directly obtaining attention scores is challenging, making implementation in existing inference frameworks problematic.

We anticipate that superior methods for identifying important KV caches may emerge. Therefore, we have maintained this as an interface (filter rules in Figure 3) in our implementation, allowing for the integration of new filters in future research.

By preserving a small portion of tokens at high precision, we achieve substantial performance gains in long-context tasks while incurring minimal additional overhead. The impact of window size on quantization performance will be discussed in Section 4.3.

# 4    Experiments

In this section, we introduce the detailed experimental settings and evaluate the effectiveness of the proposed SKVQ.

## 4.1    Settings

**Models.** We select a wide range of models with different architectures and different size to demonstrate the generalizability of our approach: Llama2-13b (Touvron et al., 2023), and models fine-tuned based on Llama2: Llama2-7b-chat, Llama2-13b-chat, Llama2-7b-80k (Fu et al., 2024), Vicuna-v1.5-7b-16k (Chiang et al., 2023), LongChat-v1.5-32k (Li et al., 2023). We also evaluate models of Mistral family which are recently very popular: Mistral-7b-v0.1(Jiang et al., 2023), Mistral-7b-instruct-v0.2. Among these models, models of Llama family adopt multi-head attention, mistral-7b-instruct-v0.2 uses multi-query attention, and mistral-7b-v0.1 uses multi-query attention and sliding-window attention.

**Tasks.** We evaluate SKVQ mainly on long sequence tasks, as this is the scenario for which KV cache quantization is most suitable. We use LongBench (Bai et al., 2023) to evaluate on various datasets. Specifically, MultiFieldQA-zh (F1 score) is a Single-Document QA task; 2WikiMultihopQA is a Multi-Document QA task; GovReport (ROUGE score) is a Summarization task; TREC (classification score) is a Few-shot Learning task; and LCC (similarity score) and RepoBench-P (similarity score) is Code Completion task. We also tested SKVQ on Needle-in-a-Haystack (Kamradt, 2023), which is a popular test-bed for whether models can actually utilize long context length. It requires the model to recite the information in a given sentence, which is placed anywhere in a long document. Finally, to provide a clearer picture of the effects of the SKVQ components and to compare with previous methods, we also measure the perplexity on wikitext2 (Merity et al., 2016) in Section 4.3.

| Model | Method | LCC↑ | RepoBench-P↑ | PR-en↑ | TREC↑ | 2wikimqa↑ | GovReport↑ | MQA-zh↑ | Average↑ |
|---|---|---|---|---|---|---|---|---|---|
| Llama-2-7B-chat | FP16 | 52.33 | 44.05 | 10.25 | 63 | 32.09 | 27.29 | 11.39 | 38.50 |
| | RTN | 15.44 | 8.76 | 0.79 | 4.00 | 0.30 | 1.93 | 0.07 | 6.76 |
| | SmoothQuant | 35.31 | 32.18 | 0.79 | 28.75 | 7.45 | 11.83 | 1.68 | 21.92 |
| | RPTQ | 22.37 | 19.08 | 5 | 47.5 | 15.57 | 20.07 | 3.24 | 19.50 |
| | KIVI | 49.32 | 43.71 | 4.50 | __63__ | 24.07 | 24.73 | 10.24 | 35.91 |
| | **SKVQ** | __50.69__ | __45.4__ | __5.5__ | 63 | __28.5__ | __27.07__ | __10.7__ | __37.50__ |
| Llama-2-13B-chat | FP16 | 50.54 | 52.1 | 15.25 | 68.5 | 13.21 | 27.52 | 7.23 | 38.83 |
| | RTN | 20.89 | 18.62 | 0.33 | 0 | 0.52 | 1.68 | 0.16 | 10.15 |
| | SmoothQuant | 32.17 | 33.86 | 2.65 | 48 | 3.53 | 12.47 | 0.47 | 23.22 |
| | RPTQ | 49.18 | 47.63 | 5.25 | 63.5 | 10.92 | 23.83 | 4.54 | 35.01 |
| | KIVI | 48.6 | 48.81 | __13.5__ | __68__ | __14.32__ | 25.7 | __7.01__ | 37.21 |
| | **SKVQ** | __49.53__ | __49.76__ | 12.25 | 67.5 | 14.03 | __26.68__ | 6.63 | __37.53__ |
| Mistral-7B | FP16 | 68.06 | 60.46 | 17.71 | 68 | 10.87 | 20.09 | 17.1 | 45.51 |
| | RTN | 27.98 | 26.18 | 3.34 | 13 | 1.11 | 2.49 | 0.45 | 15.58 |
| | SmoothQuant | 40.63 | 35.14 | 3.40 | 30.5 | 6.03 | 5 | 4.12 | 23.85 |
| | RPTQ | 55.29 | 47.12 | 5.11 | 59.5 | 9.71 | 7.81 | 12.36 | 35.05 |
| | KIVI | 65.16 | 58.33 | 12.43 | 65 | __11.03__ | 13.22 | 13.87 | 42.43 |
| | **SKVQ** | __67.81__ | __60.54__ | __13.21__ | __67__ | 10.91 | __17.72__ | __15.9__ | __43.47__ |
| Mistral-7B-Instruct | FP16 | 55.07 | 48.96 | 60 | 70 | 22.63 | 31.18 | 42.74 | 48.66 |
| | RTN | 32.36 | 33.23 | 0.67 | 1 | 2.25 | 10.03 | 2.3 | 18.02 |
| | SmoothQuant | 43.84 | 38.63 | 4.79 | 39.5 | 10.34 | 23.61 | 8.33 | 29.27 |
| | RPTQ | 46.85 | 44.07 | 27.67 | 64.5 | 16.99 | 28 | 24.68 | 38.91 |
| | KIVI | 53.13 | 48.6 | 47.5 | 69 | 20.68 | 29.37 | 33.88 | 45.48 |
| | **SKVQ** | __54.86__ | __49.05__ | __56.42__ | __70__ | __20.94__ | __30.82__ | __42.4__ | __46.23__ |

Table 1: Evaluation of different KV cache quantization methods on LongBench. Group-size(average) 128, key-cache 2bit, value-cache 2bit, window-size 128. We abbreviated PassageRetrieval as PR and MultiFieldQA as MQA. We highlight the result of our method.

**Quantization.** Both channel reorder and clipped dynamic quantization requires offline calibration. For calibration dataset, we select 256 pieces of data with length 4096 from the training set of wikitext2-v1, the calibration takes about a few minutes which is quite lightweight. We perform asymmetric quantization in all experiments. We have explored the FP8(E4M3) datatype to store scale and zero-point. Our experiment results in Table 3 show that FP8 will bring almost no performance degradation, but significantly reduces overhead at extremely low bit-width and fine grained groups.

## 4.2 Main Results and Analysis

**LongBench Results.** The performance of SKVQ in the LongBench datasets is summarised in Table 1. We compare our method with SmoothQuant (Xiao et al., 2022), RPTQ (Yuan et al., 2023) KIVI (Liu et al., 2024) and vanilla asymmetric per-token uniform RTN(Round To Nearest) quantization. SmoothQuant and RPTQ are LLM weight-activation quantization schemes. We use them to quantize KV cache without involving model weights and other activation. KIVI (Liu et al., 2024) is a recent 2-bit asymmetric quantization scheme specially designed for KV cache. We set the group size of all the methods to 128. SKVQ utilizes reordering which leads to unequal size of

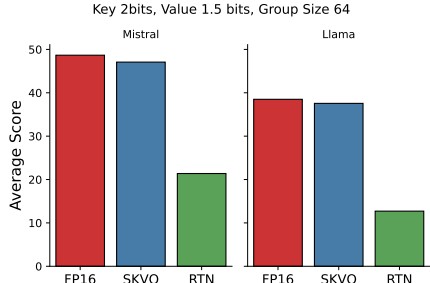

Figure 4: Average score on LongBench of SKVQ for Llama2-7b-chat and Mistral-7b-Instruct-V0.2.

each group. In order to ensure the fairness of the comparison, we control the number of groups in SKVQ to ensure the average group size is 128. The window size in SKVQ is set to 128 and the residual length in KIVI is set to 128. $\alpha$ in SmoothQuant is set to 1.0 to make the smooth transformation completely inclined to KV cache . Table 1 suggests that SKVQ is an effective method for KV cache compression that outperforms previous quantization approaches across various hard long context generation tasks. We also evaluate Vicuna-v1.5-7b-16k and LongChat-v1.5-7b-32k, the results is in Appendix 8.

For all models tested, the accuracy drop of SKVQ is less than 5%. Towards extremely low-bitwidth KV cache quantization, we further quantize the key cache into 2 bits and value cache into 1.5 bits with group size 64. The result in Figure 4 shows that SKVQ can compress key cache into 2 bits and value cache into 1.5 bits with almost no accuracy drop. It is worth noting that the experimental results are under the setting of group size 128. SKVQ can

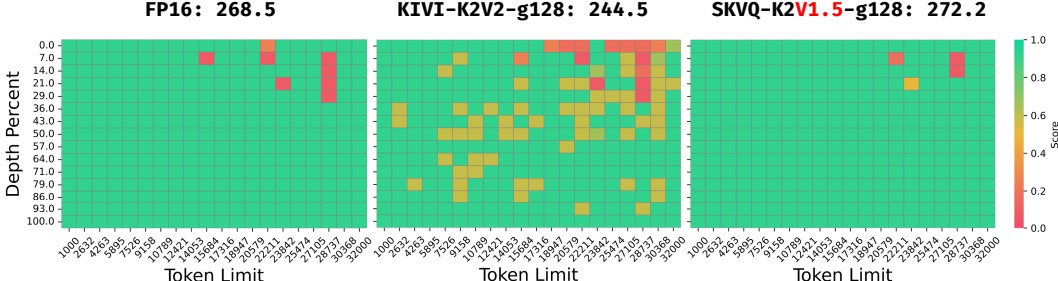

Figure 5: Comparison of SKVQ with KIVI on needle in haystack test. SKVQ achieved higher scores while using lower bitwidth.

| Method | 4bit | | 3bit | | 2bit | |
|---|---|---|---|---|---|---|
| | PPL↓ | avg-bits↓ | PPL↓ | avg-bits↓ | PPL↓ | avg-bits↓ |
| **RTN-sym** | 4.66 | 4.25 | 4.98 | 3.25 | 26.83 | 2.25 |
| **KVQuant** | 4.59 | 4.32-4.35 | 4.64 | 3.32-3.35 | 4.92 | 2.32-2.35 |
| **Ours** | 4.60 | 4.25 | 4.63 | 3.25 | 4.87 | 2.25 |

Table 2: Ablation Study: Comparison of our channel reorder based clipped dynamic quantization approach with KVQuant(best setting) and symmetric RTN per-token quantization in different quantization setting. For RTN-sym and our method, we set group-size to 64. Perplexity is Llama-2-13b test on Wikitext-2 with sequence length 4096.

also benefit from a finer-grained group, and achieve almost lossless compression, which is shown in Section 4.3.

**Needle in Haystack Results.** For needle in haystack test, we used Llama2-7b-80k (Fu et al., 2024) model for our experiments. We set the context to grow from 1k to 32k for a total of 20 intervals, and for each context length, we insert the needle into 15 different positions of the context. We compare SKVQ with KIVI under the setting of group size 128. For SKVQ, we set the window size to 128 and reserve 5 attention-sinks, i.e., when the first 5 token cache pairs slide out of the sliding window, they are retained to full precision instead of quantized to 2 bits. The residual length in KIVI is set to 128. We follow the method in (Fu et al., 2024) to calculate the recall, and finally average the scores of all test cases as the overall score. As shown in Figure 5, in key cache 2bits, value cache 2bits, group size 128 setting, KIVI got 244.5, while our SKVQ achieved 272.2 even with 2 bits key cache and 1.5 bits value cache in group size 128.

These results demonstrate that it is practical to quantize the key-value cache into extremely low-bitwidth for these tasks. More result on needle in haystack test can be found in Appendix 7.

## 4.3 Ablation Study

In this section, we decompose each part of SKVQ separately in detail and study the effect from each technique and different parameter settings.

**Breakdown of different components of SKVQ.** We investigate the accuracy impact of various quantization techniques employed in SKVQ. Initially, we utilize RTN and adopt per-token quantization with a group size of 32. Subsequently, we apply other quantization techniques used in SKVQ, including sliding window, clipping, channel reorder, attention sink, and FP8. The LongBench average scores are presented in Table 3. The attention sink size is set to 5, meaning the first 5 token cache pairs are retained at full precision. Our results indicate that sliding-window and channel re-

| Method | Avg Score↑ |
|---|---|
| FP16 | 48.66 |
| RTN | 35.55 |
| + Window-128 | 45.73 (10.18↑) |
| + Group Clipping | 46.44 (0.71↑) |
| + Channel Reorder | 47.99 (1.55↑) |
| + Attention Sink | 48.14 (0.15↑) |
| + FP8(E4M3) | 48.04 (0.1↓) |

Table 3: Ablation Study: The performance gain or loss by applying each technique in SKVQ based on RTN method. Quantization setting: kv 2bits with group size 32.

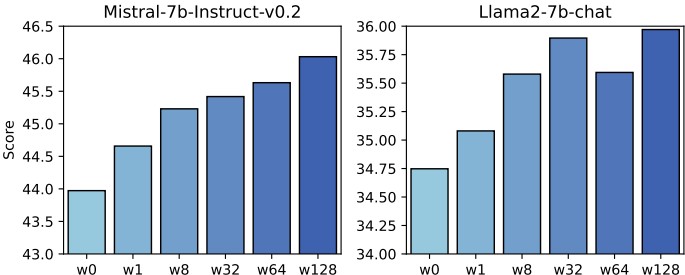

Figure 6: Ablation Study: Average score of Mistral-7b-Instruct-v0.2 on LongBench under different window sizes. Quantization setting: KV cache 2bits with group size 128

order techniques significantly enhance accuracy. FP8 (E4M3) represents the use of the FP8 datatype to store per-group quantization parameters, such as scale and zero-point. In our study, using FP8 results in a relatively minor accuracy decrease compared to FP16. However, at extremely low-bitwidth and fine-grained group sizes, using FP8 to store quantization parameters significantly reduces the average number of bits. For example, with KV 2-bit quantization and a group size of 32, using FP16 to store quantization parameters results in an average bit count of $2 + 16 * 2/32 = 3$, whereas using FP8 results in an average bit count of $2 + 8 2/32 = 2.5$, which is a 16.7% reduction.

**The effect of clipped dynamic quantization with channel reorder**. To further demonstrate the effect of our quantization approach without sliding window, we perform evaluation by measuring Llama-2-13b perplexity on wikitext2. The result in Table 2 shows that by only applying channel reorder based clipped dynamic quantization, we have outperformed KVQuant (Hooper et al., 2024). We set group size to 64 and use FP8 to store quantization parameter so that the average bits is equal to asymmetric approach adopt in ATOM and FlexGen. We also reserve the first 5 tokens to FP16 as attention sink. It worth noting that we are comparing the best score of KVQuant i.e. nuq with 1% outliers are retained to full-precision, which results in higher storage overhead than SKVQ.

**The effect of window size**. To further investigate the effect of sliding window on the final results, we set up different sizes of windows and tested them on LongBench, and the average scores are shown in the Figure 6. The result shows that the average score increases as the window size increases. In general, different sub-tasks can all benefit more or less from the sliding window strategy, and the extra overhead brought by a window with size of about 128 is negligible in long context scenarios, so we use a window of size 128 in the main experiments.

**The effect of group Size**. We vary group size from 128 to 32 to test SKVQ on LongBench, the average score on LongBench is as shown in Table 4. It shows that SKVQ can always benefit from finer-grained group. While finer-grained group brings better accuracy, it increases the computation overhead for quantization/dequantization and storage overhead for quantization parameters, which is noted as average bits. Since the performance of SKVQ on various tasks does not drop significantly when the group size is set to 128, we employ 128 group size in the main experiments.

| Group size | Avg Score↑ | Avg Bits |
|---|---|---|
| 128 | 35.365 | 2.125 |
| 64 | 35.805 | 2.25 |
| 32 | 36.51 | 2.5 |

Table 4: Ablation Study: Average scores of Mistral-7b-Instruct-v0.2 on GovReport and MultiFieldQA-zh dataset for different group sizes. Quantization setting: KV cache 2bits, window size 128.

## 5 Conclusion

In this paper, we achieve accurate ultra-low precision KV cache quantization. By channel reordering, we group similar channels together, and apply group clipping to further mitigate the outlier problem. We propose a sliding window quantization strategy with filter rules, which greatly improves the performance of the KV cache quantization method on long context tasks by reserving a small portion of the cache to full precision. By combining theses two approaches, we successfully quantize the KV cache to Key 2bits value 1.5 bits without significant precision loss. We believe this work will further advance the design of mixed-precision quantization strategies for KV cache . In the future, we will further optimize the filter rules and the kernel implementation.

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

# 6 Detailed Implementations

We describe our algorithm as shown in Algorithm 1. The subroutine get_permutation_matrix and get_group_clipping is described in Section 3.1. It's worth noting that the prologue only needs to be executed once before deploying, and we do not pay for it during the inference phase.

---

**Algorithm 1:** SKVQ Algorithm

---

**SKVQ Parameter:** window size $W$, group size $G$, filter rules $F \leftarrow \{f_1, f_2, \cdots \}$, processed KV cache length $processed \leftarrow 0$

**Attention Module Parameter:** $W_q, W_k, W_o \in \mathbb{R}^{d \times d}$

**Prologue:**
 $P_k, P_v, group\_indices \leftarrow$ get_permutation_matrix($calibration\_set$)
 $clipping \leftarrow$ get_group_clipping($calibration\_set, P, group\_indices$)
 $W_k \leftarrow P_k \cdot W_k$
 $W_v \leftarrow P_v \cdot W_v$
**end**

**Input:** $\mathbf{X} \in \mathbb{R}^{l \times d}$, $\mathbf{K_{cache}}, \mathbf{V_{cache}} \in \mathbb{R}^{h \times d}$, where $h$ is context length(prefill phase $h = 0$), $l$ is current input length(prefill phase $l = $ len($prompt$), decode phase $l = 1$)

**Algorithm** algo(*Attention module with SKVQ algorithm*)**:**
 $Q = \mathbf{X} \cdot W_q, K = \mathbf{X} \cdot W_k, V = \mathbf{X} \cdot W_v$
 $\mathbf{K_{cache}} \leftarrow$ dequant($\mathbf{K_{cache}}$)
 $\mathbf{V_{cache}} \leftarrow$ dequant($\mathbf{V_{cache}}$)
 $\mathbf{K_{cache}} \leftarrow$ concat($\mathbf{K_{cache}}, K$)
 $\mathbf{V_{cache}} \leftarrow$ concat($\mathbf{V_{cache}}, V$)
 $S \leftarrow Q \cdot$ reorder($\mathbf{K_{cache}}$)$^T$
 $O \leftarrow S \cdot$ reorder($\mathbf{V_{cache}}$) $\cdot W_o$
 $ctx\_len \leftarrow$ len($\mathbf{V_{cache}}$)
 $indices \leftarrow [processed : ctx\_len - W]$
 **if** $indices \neq \varnothing$ **then**
  $kmask \leftarrow [processed : ctx\_len - W; False]$
  $vmask \leftarrow [processed : ctx\_len - W; False]$
  **for** $filter$ in $F$ **do**
   $kmask \leftarrow filter(\mathbf{K_{cache}}[indices]) \wedge kmask$
   $vmask \leftarrow filter(\mathbf{V_{cache}}[indices]) \wedge vmask$
  **end**
  $\mathbf{K_{cache}}[indices] \leftarrow$ clipping_quant($\mathbf{K_{cache}}[indices], kmask$)
  $\mathbf{V_{cache}}[indices] \leftarrow$ clipping_quant($\mathbf{V_{cache}}[indices], vmask$)
  $processed += $ len($indices$)
 **end**
 **return O**
**end**

**function** clipping_quant($X, mask$)**:**
 $groups \leftarrow$ split_groups($X, group\_indices$)
 $quant\_cache \leftarrow \varnothing$ **for** $group$ in $groups$ **do**
  $group\_min, group\_max \leftarrow$ minmax($gruop$)
  $group\_min \leftarrow clipping[group] \times group\_min$
  $group\_max \leftarrow clipping[group] \times group\_max$
  $quant\_group \leftarrow$ group_quant($group\_min, group\_max, group$)
  $quant\_group[mask] \leftarrow group[indices]$
  $quant\_cache \leftarrow$ concat($quant\_cache, quant\_group$)
 **end**
 **return** $quant\_cache$
**end**

---

## 7 More Experimental Results of Needle in Haystack

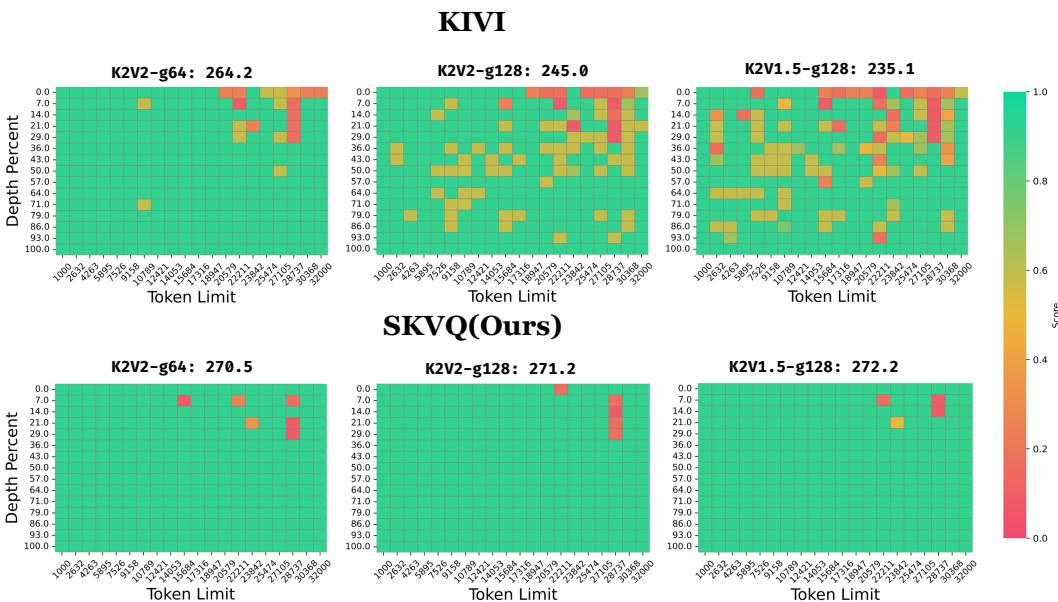

Figure 7: Comparison of SKVQ with KIVI on 32k context length needle in haystack test. The baseline score is 268.5. We vary the group size from 64 to 128, and vary the quantization bits from (key 2bits, value 2bits) to (key 2bits, value 1.5bits).

The results in Figure 7 show SKVQ is clearly better than KIVI, especially in K2V1.5-g128, SKVQ achieve the same level with FP16, while KIVI suffers significant accuracy loss. These results shows the robustness of our approach.

## 8 More Results of LongBench Evaluation

We also evaluated LongChat-v1.5-7b-32k and Vicuna-v1.5-7b-16k, which are two famous long-context models fine-tuned based on Llama2-7b. The results in Table 5 demonstrate that our SKVQ outperformed previous methods, which highlight the generalizability of our approach.

| Model | Method | LCC | RepoBench-P | PR-en | TREC | 2wikimqa | GovReport | MQA-zh | Average |
|-------|--------|-----|-------------|-------|------|----------|-----------|--------|---------|
| Vicuna-v1.5-7b-16k | FP16 | 51.38 | 46.18 | 4.5 | 69 | 21.3 | 27.79 | 43.74 | 41.02 |
| | RTN | 13.22 | 17.78 | 1.2 | 0 | 0.59 | 2.39 | 0.61 | 8.23 |
| | SmoothQuant | 40 | 29.27 | 1.94 | 18.25 | 8.33 | 14.86 | 7.19 | 22.37 |
| | RPTQ | 40.64 | 41.4 | 3.75 | 59.5 | 15.92 | 23.16 | 19.41 | 32.68 |
| | KIVI | 49.32 | 43.35 | 5.56 | 68 | 23.3 | 24.47 | 38.86 | 39.19 |
| | **SKVQ** | 50.98 | 44.07 | 6 | 69 | 22.04 | 26.55 | 40.82 | 40.20 |
| LongChat-v1.5-7b-32k | FP16 | 54.89 | 59.05 | 30.5 | 66.5 | 24.58 | 30.89 | 35.33 | 47.27 |
| | RTN | 5.11 | 3.73 | 1.5 | 0 | 0.42 | 0.51 | 0.08 | 2.461 |
| | SmoothQuant | 36.21 | 31.91 | 2.45 | 36.5 | 13.94 | 17.21 | 6.59 | 24.70 |
| | RPTQ | 40.4 | 43.2 | 8 | 61 | 17.31 | 24.79 | 20.01 | 34.01 |
| | KIVI | 49.86 | 54.77 | 20.5 | 66 | 23.79 | 28.75 | 31.58 | 43.22 |
| | **SKVQ** | 55.01 | 57.24 | 22 | 67 | 22.4 | 30.03 | 31.68 | 45.37 |

Table 5: Evaluation resultes of Vicuna and LongChat on LongBench. Group-size(average) 128, key-cache 2bit, value-cache 2bit, window-size 128. We abbreviated PassageRetrieval as PR and MultiFieldQA as MQA. We highlight the result of our method.

## 9 Memory and Latency Analysis

In order to further illustrate the benefits of quantizing KV cache to extremely low-bitwidth, we use LLM-Viewer (Yuan et al., 2024) to analyze the benefits in terms of memory consumption and inference latency. The result is shown in Table 6. When batch size and sequence

length are relatively large, KV cache dominates almost all the memory consumption, and load KV cache becomes the performance bottleneck of the entire inference system. By quantizing KV cache with SKVQ, we can significantly reduce both latency and memory consumption. The analysis result shows SKVQ enables 1M context length in a single A100-80GB. As for the inference latency, we show the results of decoding phase, which domain the inference time in long context tasks. In the case of batch size 128 and sequence length 200k, the theoretical 7x speedup can be achieved.

| Batch Size | Seq Length | Latency(ms) / Memory(GB) | FP16 | KV4 | KV2 |
|---|---|---|---|---|---|
| 1 | 32k | Inference Time
Memory Access
Memory Consumption | 10.6
21.6
29.7 | 7.5
15.3
17.2 | 7
14.3
15.1 |
| | 128k | Inference Time
Memory Access
Memory Consumption | 23.1
47.2
80.1 | 10.8
22
29.7 | 8.7
17.8
21.4 |
| | 200k | Inference Time
Memory Access
Memory Consumption | 32.5
66.3
118 | 13.3
27
39.2 | 10
20.5
26.1 |
| 64 | 32k | Inference Time
Memory Access
Memory Consumption | 274.1
559
1100 | 76.6
156
282 | 43.7
89.1
147 |
| | 128k | Inference Time
Memory Access
Memory Consumption | 1100
2200
4300 | 286.4
584
1100 | 154.8
316
551 |
| | 200k | Inference Time
Memory Access
Memory Consumption | 1700
3400
6700 | 443
905
1700 | 238.1
485
853 |
| 128 | 32k | Inference Time
Memory Access
Memory Consumption | 541.8
1100
2200 | 146.8
299
550 | 81
165
282 |
| | 128k | Inference Time
Memory Access
Memory Consumption | 2100
4400
8600 | 566.4
1200
2200 | 303.1
618
1100 |
| | 200k | Inference Time
Memory Access
Memory Consumption | 3300
6800
13400 | 881.1
1800
3400 | 469.7
958
1700 |

Table 6: LLaMA-7B memory and latency analysis with roof line model. The hardware platform is A100 80G, we assume flash-attention is used.

## 10 Comparison Between Smooth and Reorder

To improve the accuracy of per-token quantization, we utilize the reorder to cluster similar channels together. There are also other methods to improve the accuracy, one of them is smoothing, which is adopted in (Xiao et al., 2022; Shao et al., 2023; Yue et al., 2024). This approach smooth the difference between channels by multiplying an extra factor before quantization. We explore and compare smoothing with reordering, the experimental results are shown in Table 7. SKVQ-smooth represents our sliding window strategy together with smoothing and SKVQ-reorder represents the approach we described in Section 3.1. The results demonstrate that reordering can effectively improve the per-token quantization performance while the smoothing cannot. This is mainly because smoothing does not take into account the differences in token dimensions.

| Model | Method | LCC | RepoBench-P | PR-en | TREC | 2wikimqa | GovReport | MQA-zh | Average |
|---|---|---|---|---|---|---|---|---|---|
| | FP16 | 52.33 | 44.05 | 10.25 | 63 | 32.09 | 27.29 | 11.39 | 38.50 |
| LLaMA-2-7B-chat | SKVQ-reorder | 50.69 | 45.4 | 5.5 | 63 | 28.5 | 27.07 | 10.7 | 37.50 |
| | SKVQ-smooth | 48.93 | 40.12 | 4.75 | 62.5 | 26.75 | 23.19 | 7.93 | 34.77 |
| | FP16 | 50.54 | 52.1 | 15.25 | 68.5 | 13.21 | 27.52 | 7.23 | 38.83 |
| LLaMA-2-13B-chat | SKVQ-reorder | 49.53 | 49.76 | 12.25 | 67.5 | 14.03 | 26.68 | 6.63 | 37.53 |
| | SKVQ-smooth | 47.78 | 47.28 | 7.5 | 67 | 11.61 | 24.07 | 5.55 | 35.34 |
| | FP16 | 68.06 | 60.46 | 17.71 | 68 | 10.87 | 20.09 | 17.1 | 45.51 |
| Mistral-7B | SKVQ-reorder | 67.81 | 60.54 | 13.21 | 67 | 10.91 | 17.72 | 15.9 | 43.47 |
| | SKVQ-smooth | 64.18 | 57.95 | 9.49 | 63.5 | 10.11 | 13.99 | 12.77 | 41.52 |
| | FP16 | 55.07 | 48.96 | 60 | 70 | 22.63 | 31.18 | 42.74 | 48.66 |
| Mistral-7B-Instruct | SKVQ-reorder | 54.86 | 49.05 | 56.42 | 70 | 20.94 | 30.82 | 42.4 | 46.23 |
| | SKVQ-smooth | 49.83 | 45.74 | 40.42 | 66 | 17.11 | 28.32 | 30.32 | 42.11 |

Table 7: Comparison of different methods(i.e. smooth v.s. reorder) on LongBench. Group-size(average) is set as 128, key-cache 2bit, value-cache 2bit, window-size 128. We abbreviated PassageRetrieval as PR and MultiFieldQA as MQA.

