# OpenReview forum: "SKVQ: Sliding-window Key and Value Cache Quantization for Large Language Models"
_colmweb.org/COLM/2024/Conference — COLM_

### Official Review · Reviewer_KwiV · 2024-05-08

**Rating:** 7
**Confidence:** 4
**Ethics Flag:** 1

**Summary:**

This paper proposes a KV Quantization method to address the memory overhead encountered when processing long contexts in LLMs. The method, called SKVQ, combines traditional KV-cache compression strategy via a sliding window with new quantization techniques like channel re-ordering and group clipping. This approach demonstrates superior performance on LongBench, Needle in a Haystack, and PPL benchmarks compared to other KV Quantization methods (e.g., KIVI, KVQuant), even at low bit levels such as 2-bit or 1.5-bit.

**Questions To Authors:**

- **Start Token Retention in Attention Sink**: Is the number of start tokens retained determined heuristically, or was there a specific criterion used?
- **Overhead in Prefill Stage for Very Long Contexts**: Considering scenarios like NarrativeQA in LongBench with extremely long input contexts (over 80K), was there any notable overhead during the prefill stage when using FP to calculate KV embeddings? It seems likely that memory consumption would be significantly high in these very long contexts.
- **Effectiveness of H2O**: It was mentioned that the impact of H2O was minimal. Could you clarify this numerically?
- **1.5-bit Quantization**: Does the 1.5-bit specification imply the application of ternary quantization?
- **Positional Encoding and Quantization**: There was no mention of Rotational Positional Encoding (RoPE). When considering RoPE's rotating nature, is the quantization applied pre-RoPE like in KVQuant, or is channel re-order applied after RoPE has been integrated? This missing information is crucial for understanding the interaction between RoPE and the quantization process.
- **Quantization Kernel and Memory Storage Details**: You mentioned storing quantization parameters in FP8. Are you using a specific quantization kernel for this, and when storing KV cache at 2-bit and 1.5-bit, is fake quantization employed? More details on the kernel implementation would be informative, especially regarding how memory efficiency is achieved.

**Reasons To Accept:**

- Despite being a KV Quantization method, the sliding window quantization framework opens opportunities to incorporate insights from existing KV cache eviction methods (e.g., H2O), providing a novel and impressive aspect.
- The evaluation in long-context scenarios, such as LongBench, (not just MMLU, CSQA, ...) where KV optimization is critical, makes this paper a relevant contribution for KV cache compression research.
- Detailed ablation studies (examining channel re-order and group clipping effects, window size sweep experiments, and task-specific analyses) reinforce the method's effectiveness and provide additional insights to readers.

**Reasons To Reject:**

- The evaluation would have benefited from testing long generation scenarios in addition to long-context scenarios. During the prefill stage, FP is used to calculate the KV Cache, making KV embeddings relatively robust to quantization. However, as the generation length increases, the impact of quantization on KV embeddings created during the generation stage remains unexplored in this paper. It would have been beneficial to assess whether SKVQ remains robust in scenarios where LLMs generate long texts, such as agent-based conversations.
- The method has limitations in terms of novelty. The sliding window approach (from KV cache eviction works), channel re-ordering (from KVQuant), and group clipping (from AWQ) are all derived from existing methodologies. This reduces the originality of the SKVQ method.

---

> ### Author Rebuttal · Authors · 2024-05-31
>
> > **Q1**:  Evaluation on long generation scenarios
>
> Our space is very limited, please refer to the reply to `Q2` of `QPwH`
>
> > **Q2**: novelty
>
> - sliding window: We are the first to **thoroughly exploring the potential of attention locality in KV-cache quantization** and designing an algorithm that seamlessly fuses it with KV-cache quantization.
>
> - AWQ use auto-clip for weight quantization. Our research discovered that **approximating and precomputing clip thresholds on a small calibration set effectively addresses the clipping challenge** for KV-cache and effectively improves accuracy.
>
> - KVQuant does not employ channel reordering. SKVQ conducts **per-token uniform quantization** by extracting value distribution features from each channel.
>
> > **Q3**: Start Token Retention in Attention Sink
>
> The value here is usually an empirical setting. In our experiments, we set this number to `5`(which is `4` in streaming-llm).
>
> > **Q4**: Overhead in Prefill Stage for Very Long Contexts
>
> Our space is very limited, please refer to the refer to the reply to `Q1` of `QPwH`
>
> > **Q5**: Effectiveness of H2O
> There are bunch of details to implement heavy hitter detection in our framework. But the rebuttal space is limit to 2500. We look forward to continuing our discussion in discussion period.
>
> Here are some basic results. **K3V2** setting, group size set to 32, window 32, `H2O` represent enabling the heavy hitters in filter rules, while `Random` represent randomly retain some token cache to full-precision. We set the ratio of full precision to **4%**.
>
> | Setting | RTN | H2O | Random |
> | ------- | ------------------ | ------ | ------ |
> | K3V2    | 257.41 | 261.58 | 261.91 |
>
> > **Q6**: RoPE and Quantization? ternary quantization?
>
> 1.5bits means ternary quantization. We perform pre-RoPE quantization:
>
> - Quantization Step:  QKV_projection $\Rightarrow$ channel-reorder $\Rightarrow$ quantize $\Rightarrow$ store together with `past_kv`
>
> - Decoding Step: load quantized KV $\Rightarrow$ dequant $\Rightarrow$ channel-reorder $\Rightarrow$ RoPE $\Rightarrow$ Attention
>
> > **Q8**: fake quantization? Kernel and Memory Storage Details
>
> All the results are obtained **through fake quantization**.
> we do have custom kernel, there is no more space to describe it in detail, we look forward to disccusion in discussion period. Basically, we adjust the output channels of the weigh to avoid explicit reordering, pack every four adjacent numbers within the same group into one `uint8` for storage.

---

> > ### Author Response · Authors · 2024-05-31
> > **supplement to the rebuttal**
> >
> > > **Q2**: novelty
> >
> > First, we want to clarify that the problem we are addressing is different from the issues tackled by existing methods. Our primary focus is on ultra-low-bit quantization of KV-cache. Our method is designed based on the key insight
> >
> > 1. we identified outlier channels in the key cache and innovatively recognized that the locality of the attention module can be combined with mixed-precision quantization.
> >
> > 2. We hypothesize that preserving a small portion of the KV-cache (designed as a sliding window) in high precision can effectively reduce quantization error, and our detailed experimental analysis supports this hypothesis.
> >
> > While some techniques have been employed in various domains before, integrating them to minimize KV-cache quantization error is not trivial. For example, our experiments compared smoothquant and RPTQ, which utilize smoothing and channel-reordering, respectively, and demonstrated that both perform poorly in ultra-low-bit quantization of KV-cache.
> >
> > Regarding the three techniques you mentioned, we would like to highlight the differences from previous work:
> >
> > 1. Designs similar to the sliding-window have been extensively explored in approximate linear attention models like Longformer[1], and the structural property of attention, particularly locality, has been well-studied in previous works. These studies inspired us and provided a solid theoretical foundation for our method. However, no prior work has combined quantization with the inherent structured nature of attention. We are the first to integrate these two aspects, **thoroughly exploring the potential of attention locality in KV-cache quantization** and designing an algorithm that seamlessly fuses it with KV-cache quantization.
> >
> > 2. While auto-clip techniques have been applied in works like AWQ[2], they were used for weight quantization. We are the first to apply clipping techniques to KV-cache quantization. There is a significant difference between the two: since weight distributions are known, precomputing clip values offline is straightforward. In contrast, KV-cache is generated dynamically, making online auto-clipping in dynamic quantization impractical due to unacceptable overhead. Our research discovered that **approximating and precomputing clip thresholds on a small calibration set effectively addresses the clipping challenge** for KV-cache and effectively improves accuracy.
> >
> > 3. KVQuant does not employ channel reordering; it performs per-channel non-uniform quantization by clustering within each channel. In contrast, SKVQ conducts **per-token uniform quantization** by extracting value distribution features from each channel, grouping similar channels, and applying uniform quantization to each group. The concept of channel reordering has indeed been used in RPTQ[3], but [3] aims to reduce weight-activation quantization loss through reorder-indexing. We firstly apply channel-reordering to KV-cache quantization.
> >
> > We will further emphasize our insights and clarify the differences between our approach and previous work in the revised version. Thank you for pointing it out!
> >
> > [1] Beltagy, Iz, Matthew E. Peters, and Arman Cohan. "Longformer: The long-document transformer." *arXiv preprint arXiv:2004.05150* (2020).
> >
> > [2] Lin, Ji, et al. "Awq: Activation-aware weight quantization for llm compression and acceleration." *arXiv preprint arXiv:2306.00978* (2023).
> >
> > [3] Yuan, Zhihang, et al. "Rptq: Reorder-based post-training quantization for large language models." *arXiv preprint arXiv:2304.01089* (2023).

---

> > ### Author Response · Authors · 2024-05-31
> > **supplement to the rebuttal**
> >
> > > **Q5**: Effectiveness of H2O
> >
> > Firstly, we'd like to clarify that while H2O serves as a cache eviction policy, it isn't directly applicable to quantization in KV-cache. Achieving the goal of using accumulative attention scores to determine the important token for mixed-precision quantization requires careful design.
> >
> > There are a lot of details, let's begin from vanilla dynamic per-token RTN quantization during decoding. We face a dilemma: on one hand, we need to generate more tokens to assess the importance of the current token; on the other hand, we need to make an immediate decision whether to quantize this token to a low-precision.
> >
> > A sliding-window strategy resolves this effectively! Upon generating a token pair `cache_t` at decoding step $t$, we defer quantization until the it slides out of the window (i.e., at decoding step $t+W$). Then, based on the accumulative attention scores of the subsequent $W$ tokens on `token_t`, we decide whether to retain it at high precision.
> >
> > Our strategy is implemented as follows: for a token $t_i$ that slides out of the window, we accumulate it's attention score in $step_{i+1},\cdots, step_{i+W}$. If this score exceeds the cumulative score of the previous $96%$(or other threshold) of tokens, we consider this token as a **heavy hitter** and protect it from quantizing to low-precision.
> >
> > The score in the table is the average score in needle in haystack test, same with Figure5 in our paper. We set the ratio of full precision to **4%**. we use RTN, without channel reorder and clipping.
> >
> > The results indicate that enabling heavy hitter in filter rules marginally improves accuracy. It can't outperform the random baseline, indicating that our current design approach does not yield significant benefits. At the same time, it comes with 2 limitations:
> >
> > 1. Heavy hitter determination relies on attention scores, which are inaccessible during inference with existing fusion operators like flashattention, flashinfer. This necessitates a custom operator for accessing cumulative scores, leading to significant overhead.
> > 2. The dynamic nature of heavy hitters requires an additional index at runtime, introducing substantial overhead, particularly in ultra-low-bit quantization.
> >
> > These drawbacks hinder the feasibility of this approach. Thus, while we haven't further adopted this strategy in our experiments, we believe there are more efficient ways to implement heavy hitters detection and more effective filter rules to enhance SKVQ performance.

---

> > ### Author Response · Authors · 2024-05-31
> > **supplement to the rebuttal**
> >
> > > **Q6**: RoPE and Quantization?
> >
> > Because we perform pre-RoPE quantization, we store pre-RoPE KVCache, requiring RoPE to be applied at each decoding step. Experiments in FlashInfer[1] has shown Fused-RoPE Attention kernel can apply RoPE on the fly with negligible overhead.
> >
> > [1] Ye, Z. FlashInfer: Kernel Library for LLM Serving.
> >
> > > **Q8**: Is fake quantization employed? Quantization Kernel and Memory Storage Details
> >
> > Regarding the details of FP8: We simulate the use of FP8 by first computing the quantization parameters (zero-point and scale) in FP16, then converting the values to FP8 using `round-to-nearest-even` rounding, and finally converting them back to FP16.
> >
> > kernel implementation details: We have currently implemented the quant/dequant CUDA kernel and are continuing to optimize it. Specifically, as mentioned in the answer to Q5, the kernel details for the quantization process are as follows:
> >
> > 1. We adjust the output channels of the weight (`QKV_projection`) to avoid explicit reordering.
> > 2. Using the pre-computed reorder index, we determine the range of each group and calculate the group-shared scale and zero-point in FP16, then convert them to FP8, store in sequence by group.
> > 3. Next, we perform packing: for the 2-bits K/V cache, we pack every four adjacent numbers within the same group into one `uint8` for storage. Although the group size might not align perfectly with 4, leading to some waste, this is negligible (at most $0.8 \times NumGroup$ bytes).
> >
> > After quantization and packing, the original KV-cache shape $[bs, SeqLen, NumHeads, HeadDim]\Rightarrow [bs, SeqLen, PackDim]$, where $PackDim \approx NumHeads \times HeadDim / 8$. This reduces the memory footprint of the KV-cache by around 800% compared to FP16.
> >
> > During the decoding step:
> >
> > 1. Based on the starting position index of the packed groups (obtained from the reorder index during the offline pre-computation), we determine the starting position of each group, find the corresponding scale and zero-point, and dequantize to FP16. Then, we reorder the results back to the original sequence using the inverse reorder index and write them back to HBM.
> > 2. Execute RoPE and the Attention Kernel with FP16 KV-cache.
> >
> > It worth noting that the dequant, subsequent RoPE, and Decoding Kernel can be fused together, eliminating the need to write the intermediate dequantized KV-cache back to HBM. We are continuously optimizing this fused kernel, and will open source it.

---

> > > ### Comment · Reviewer_KwiV · 2024-06-05
> > >
> > > Thank you for your detailed response. My concerns have been addressed. I also acknowledge the difficulties in incorporating the H2O methodology. Also, I'd like to suggest that maybe you can apply the SnapKV[1] methodology alongside FlashAttention. I would appreciate if the revised version includes experimental results on long context generation thoroughly.
> > >
> > > Happy to increase my score to 7!
> > >
> > > [1] Li et al, SnapKV: LLM Knows What You are Looking for Before Generation, 2024 Arxiv

---

> > > > ### Author Response · Authors · 2024-06-07
> > > > **Thanks**
> > > >
> > > > We sincerely appreciate your thorough and conscientious review. We are very pleased to receive your insightful feedback. Thank you once again.

---

### Official Review · Reviewer_nGVZ · 2024-05-09

**Rating:** 7
**Confidence:** 3
**Ethics Flag:** 1

**Summary:**

This paper proposes SKVQ, a version of KV-cache quantization that utilizes two further ideas to improve quality and efficiency: a recent-token-keeping mechanism and a channel reordering schema coupled with dynamic quantization. Each contribution is evaluated in ablation studies and in a fitting setup, with the conclusion being that SKVQ outperforms existing methods and should be adopted for LLM generation.

**Questions To Authors:**

### Questions
* What value of K did you choose for KMeans (p.4 center)? Did this choice make a difference?
* Why do you claim to have outperformed KVQuant in p.8 when Table 2 shows otherwise?
* What do the colors in Figure 5 signify? There's no legend.

### Required fixes
* The figures require color display to make out, especially 1, 5, and 6 (colors being referenced from the text). This is inconsiderate towards colorblind folks and anybody printing the paper grayscale. Please make the necessary adjustments.
* Some notions are mentioned without proper introduction, for example "locality of attention" in page 2 (only elucidated in p.5)
* h is defined twice, in eq. 2 and just several lines above, for no apparent reason
* The choice to highlight the performance of your own method in the tables, as opposed to bolding teh best-performing methd, is misleading. Your method has its own row and doesn't need highlighting. Please revert to conventional practice.

### Grammar
As mentioned above, there are many grammatical errors in the paper, mostly with respect to determiner placement ("a" / "an" / "the") but also verb-subject number agreement, some peculiar phrasings ("combined together"; "a chance" where "an opportunity" is meant), and hyphenation.
There are also many spacing issues, mostly missing spaces before parentheses (especially those of citations).

Foci of errors include:
* the abstract
* the last two paragraphs of the introduction
* the first two of section 2
* the first of 3.1
* "we perform..." on p.4
* footnote 2
* most of p.5, until the end of section 3
* first sequence of 4.3 (which is not a sentence)
* the next three paragraphs (what does "adopt in ATOM" mean?)
* Figure 6's description
* "the effect of group size" paragraph

Just go carefully over the whole thing, please.

**Reasons To Accept:**

* The motivation is clear, the idea is sound, and the experiments are convincing.
* The argument structure of the paper is good.
* The amount of technical detail in section 3.1 is just right.

**Reasons To Reject:**

Nothing that merits rejection, but there are **a lot** of grammatical errors in this paper. Frustratingly enough, they are localized to (a large number of) specific paragraphs (see below), which raises the concern that at least one of the authors has sufficient competence in English and could have easily made a pass over the malformed paragraphs but didn't bother. This is upsetting as it is not the reviewers' role to serve as copyeditors (and it distracts from evaluating the content).

---

> ### Author Rebuttal · Authors · 2024-05-31
>
> Thanks for your thoughtful review that will help us strengthen the manuscript.
>
> > **Q1**: What value of K did you choose for KMeans (p.4 center)? Did this choice make a difference?
>
> Thank you for pointing this out, we will explain this part in detail in the revised paper.
>
> To ensure a fair comparison, we set $K$ to ensure that the average group size for SKVQ matches the group size in other methods.
> For example, with Llama2-7b-chat with **MHA** ($\text{NumKVHeads} =\text{NumHeads} = 32$, $\text{HeadDim} = 128$), for the main experiment setting of "group-size 128", we set $K = 32$. Hence, total $\text{NumKVHeads}\times\text{HeadDim}$ channels are divided into 32 clusters, each considered as a quantization group. Although each group size varies, the average group size is $\text{NumHeads} \times \text{HeadDim} / K = 128$.
>
> > **Q2**: Why do you claim to have outperformed KVQuant in p.8 when Table 2 shows otherwise?
>
> Please note that Table 2 on page 8 reports the model **perplexity(PPL)** obtained on Wikitext-2, where **lower values are better**. In the 4-bit setting, `Ours` achieves a PPL of $4.60$ while `KVQuant` achieves $4.59$, which is slightly better. However, it worth noting that the average bits of `KVQuant` is also larger than `Ours` due to the storage overhead caused by outlier index.
>
> When it comes to 3-bits and 2-bits, `Ours` achieved a lower PPL than `KVQuant` while maintaining lower average bits.
>
> > **Q3**: What do the colors in Figure 5 signify? There's no legend.
>
> We sincerely apologize and appreciate you pointing out this issue. colors closer to green indicate higher scores for the model at that test point, while colors closer to red indicate lower scores.
>
> We overlooked the legend when creating the plot. We acknowledge that the current figure is not friendly to colorblind readers. We will reconsider the result display in the revised paper and add appropriate text explanations. Thank you for your valuable suggestion.
>
> > **Q4**: Grammar issues and other bugs in paper.
>
> We sincerely apologize for the inconvenience caused by these detailed errors. We agree that authors should ensure grammatical errors do not distract reviewers. We will correct the careless mistakes you pointed out and carefully recheck each paragraph.
>
> Thank you for your meticulous feedback again!

---

### Official Review · Reviewer_MMd8 · 2024-05-10

**Rating:** 7
**Confidence:** 3
**Ethics Flag:** 1

**Summary:**

The paper presents a new variant of KV Cache compression for LLMs.

**Questions To Authors:**

There are certain aspects in the manuscript that need to be improved.

"RTN" is mentioned in tables (first time on page 7) but is not defined/explained in the text.
It is somewhat defined in the legend of table 2, but needs to be explained earlier in the manuscript, and in more detail.

Table 1 presents evaluation, but what are the numbers that we see in it? Are those miles per hour, or salary per day? Or scores (percents) on the tasks?  Please care to define what are the values we see (what are the units) and how the scale should be interpreted (higher or lower values are better?).
Also mention that in the relevant part of the text.

Grammar issues:
The word 'but' is used in several places where it is not grammatical. e.g. page 5: "large amount of but less important content", "small portion of but more important caches".
Another error, page 2: "we discover that the protecting the accuracy" (drop 'the' after 'that').

Page 3: "these methods do not take into account the differences in token dimensions."
This is not quite clear. If you are talking about text tokens, their dimensions are fixed via embeddings dimension. So what tokens are you talking about and how do their dimensions differ?  This should be clarified.

This publication needs to be cited, as it as also about sliding window  approach:
https://arxiv.org/abs/2401.01325.

Page 8 (bottom of page):
"cache is 2 + 16 ∗ 2/32 = 3, but if we use FP8, then average bits is 2 + 8 ∗ 2/32 = 2.5, which is 50% smaller."
This is incorrect, 2.5 is not 50% smaller than 3. it is only 17% smaller than 3.

**Reasons To Accept:**

The presented compression idea is neat, and the evaluations show promising results. Ablation studies are presented that decompose the proposed approach and show the influence of its components.

**Reasons To Reject:**

The paper is very condensed and would not be easy to understand for wider NLP audiences. However, for those working on KV caches, it is adequate.

A major claim in the manuscript,
"we further quantize the key cache into 2 bits and value cache into 1.5 bits"
is not clear at all.
It is not explained in the theory part, and is mentioned as given in the experiment part.
This claim should be explained in detail, showing how you arrive at this estimation.

---

> ### Author Rebuttal · Authors · 2024-05-31
>
> Thank you for your suggestions on our manuscript. We find most of them to be very accurate and appreciate your careful attention. We believe your insights will significantly enhance the quality of our paper. Sincere thanks!
>
> > **Q1**: estimation about quantizing the key cache into 2 bits and value cache into 1.5 bits.
>
> Thank you for highlighting this issue. In general, our estimates are based on two aspects:
>
> Firstly, we noticed that the accuracy of `KIVI`(previous SOTA) at K2V2 significantly decrease as the group-size increasing. We think this means that `KIVI` does not handle outliers within a group well. This presents an opportunity for superior performance.
>
> Secondly, our approach is driven by experimental results. As shown in Table 1 of the manuscript, with a group size of 128, key cache at 2 bits, and value cache at 2 bits, the SKVQ scores were very close to FP16. This motivated us to explore lower precision.
>
> We will emphasize this process in the revised version of the paper. Thank you for pointing it out.
>
> > **Q2**: "RTN" in page 7 is unclear.
>
> You are absolutely right; we did not explain "RTN" in detail in the original text. In fact, apart from "RTN-sym" in Table 2, "RTN" in our experiments refers to asymmetric per-token group quantization, which applies the round-to-nearest strategy.
>
> We will clarify this in the revised manuscript. Thank you for pointing it out!
>
> > **Q3**: Grammar issues and other error
>
> We sincerely appreciate your careful review and the identification of grammatical errors. We will thoroughly review the manuscript and correct these issues to improve clarity and readability. Thank you for bringing these to our attention.
>
> > **Q4**: confusion about the  phrase "differences in token dimensions"
>
> Here is a revised, clearer version:
>
> we use the term "token dimension" to refer to the `seq_len` dimension in KV-cache. Specifically, for KV-cache with shape $[\text{bs}, \text{SeqLen}, \text{NumHeads}, \text{HeadDim}]$, we use "token dimension" to indicate the second dimension($\text{SeqLen}$).
>
> We observed that, in addition to fixed outlier channels which is observed in previous work, there is also variance between different tokens. Simply applying channel-wise smoothing does not address the issues in this dimension. To resolve this, we propose Clipped Dynamic Quantization.
>
> We apologize for any confusion caused by our original wording, and will correct it in revised version.

---

> > ### Comment · Reviewer_MMd8 · 2024-05-31
> > **concerning 2 bits and 1.5 bits**
> >
> > I am satisfied with author replies.
> >
> > About 2 bits and 1.5 bits.
> > I understand that in this paper, those are just statistical averages.
> > Yet I want to emphasize that
> > with 2 bits, one can distinguish just 4 values.
> > And with 1.5 bits - only '3' (of course practically one cannot encode half a bit).
> > But it all means that after quantization you end up with a very small repertoire of values.
> > It looks like those values are close to acting as yes/no gates on the attention.
> > This is speculative, but maybe the authors want to consider this.

---

> > > ### Author Response · Authors · 2024-06-01
> > > **concerning 2 bits and 1.5 bits**
> > >
> > > Your speculation is largely correct. After ternary quantization(1.5 bits quantization), the values within a group are quantized to {-1, 0, 1}, making the dot-product operation (as used in attention computation) entirely a select-and-accumulate operation, just as you described with the "Yes/No" gate analogy.
> > >
> > > However, we must point out that since SKVQ performs asymmetric uniform quantization with group, after the selection and accumulation through the "Yes/No" gate, scaling is required, and the zero-point compensation must be added.
> > >
> > > You have provided an excellent perspective, which is very intriguing. If you have any further ideas, we would be delighted to discuss them with you further.

---

### Official Review · Reviewer_QPwH · 2024-05-10

**Rating:** 7
**Confidence:** 4
**Ethics Flag:** 1

**Summary:**

The paper introduces a novel method, SKVQ (Sliding-window Key and Value Cache Quantization), that mitigates the memory consumption issue caused by the key-value (KV) cache in large language models (LLMs). The authors propose an extremely low bitwidth quantization strategy that deploys channel reordering, clipped dynamic quantization at group granularity, and a sliding window approach to ensure data accuracy while achieving high compression ratios. The experimental results demonstrate significant improvements over existing methods with minimal loss of precision.

**Questions To Authors:**

How would it perform on larger models, e.g. llama-2-70b? Possibly together with tensor parallelism? It would be beneficial for future revisions to consider evaluating the proposed methods on larger models.

**Reasons To Accept:**

1. This paper tackles an important problem in deploying large language models - the memory consumption of KV caches. And the paper combines several techniques such as channel reordering and clipped dynamic quantization to minimize the quality loss.
2. The evaluation is extensive and relatively thorough. The result shows that SKVQ surpasses previous approaches in terms of accuracy under different average bit widths.
3. The paper is well-written and provides in-depth insights into the proposed method.

**Reasons To Reject:**

1. Since the prefill stage is computed in full-precision, it could become a bottleneck especially when the generation length is short compared to the context length.
2. It would be great if the proposed method can be evaluated in long generation scenarios, which may present different challenges than long-context scenarios.

---

> ### Author Rebuttal · Authors · 2024-05-31
>
> > **Q1**: The full-precision prefill stage can bottleneck when the prompt length is long.
>
> Quantization is executed layer-by-layer, not all KV-cache stay in full-precision in prefill phase. Instead, there is only 1 layer's KV-cache in full-precision.
>
> Assuming prompt length is 100k,$bs=1$, for Llama2-7B with $\text{HeadDim} = 128$,$\text{NumHeads} = 32$, peak memory footprint in prefill phase:
>
> - FP16: $2 \times 100000 \times 2 \times 32 \times 32 \times 128 = 50 \text{GB}$
>
> - SKVQ: $2 \times 100000 \times \left(\frac{2 \times 31 \times 32 \times 128}{9.14} + 2 \times 1 \times 32 \times 128\right) = 7.4 \text{GB}$
>
>
> Therefore, SKVQ also **achieves a high compression rate at prefill phase**.
>
> > **Q2**:  Evaluation on long generation scenarios
>
> First, there is not suitable dataset for long generation task.
> To address your concerns, we manually created some prompts with instructions to generate as long as possible (at least 3000 words). We test on Llama2-7b-chat. For each prompt, we generated responses under three settings: full precision, SKVQ, and KIVI. The SKVQ and KIVI models were quantized with K 2bits and V 1.5bits, group size set to 128, window-size for SKVQ at 128, and residual length for KIVI at 128.
>
> We utilize GPT4-o, Gemini-1.0-Pro to judge the responses.
>
> The data in the table represents the score of **FP16/SKVQ/KIVI** respectively.
>
> | prompt | GPT4-o   | Gemini-1.0-Pro | Claude-Sonnet |
> | ------ | -------- | -------------- | ------------- |
> | p1     | 30/34/22 | 32/40/24       | 30/36/27      |
> | p2     | 35/30/33 | 36/39/36       | 30/38/36      |
> | p3     | 26/32/19 | 18/37/30       | 9/35/22       |
>
> > **Q3**: Evaluation  on larger models?
>
> Evaluation on Llama3-70B-Instruct.
>
> |                      | FP16  | SKVQ  | Relative to FP16 |
> | -------------------- | ----- | ----- | ---------------- |
> | RepoBench-P          | 68.78 | 67.69 | 98.42%           |
> | passage_retrieval_en | 68.83 | 68.33 | 99.27%           |
> | multifieldqa_en      | 18.52 | 18.14 | 97.95%           |
> | Qasper               | 12.2  | 11.46 | 93.93%           |
> | TREC                 | 73.5  | 73.5  | 100.00%          |
> | 2wikimqa             | 18.37 | 18.05 | 98.26%           |
> | gov_report           | 32.82 | 31.96 | 97.38%           |
> | multifieldqa_zh      | 20.1  | 20.02 | 99.60%           |
> | multi_news           | 26.83 | 26.9  | 100.26%          |
> | qmsum                | 22.25 | 21.88 | 98.34%           |
> | samsum               | 44.41 | 44.78 | 100.83%          |

---

> > ### Author Response · Authors · 2024-05-31
> > **supplement**
> >
> > > **Q1**: The full-precision prefill stage can bottleneck when the prompt length is long.
> >
> > Since we are not sure whether your concern about the "bottleneck" at prefill phase refers to **computation** or **memory**, in addition to the above explanation of memory, we will add an additional explanation of computation in prefill phase:
> >
> > For each layer, we use the full-precision activation generated from the `QKV_projection` for attention computation, which is identical to the full-precision model. Additionally, we quantize the full-precision KV embedding into low-precision KV-cache for storage, which introduce a lightweight overhead.
> >
> > Using full-precision KV embeddings for attention at prefill phase is a common practice in many quantization algorithms, such as FlexGen and KIVI. Overall, SKVQ provides **no speedup compared to full-precision model at prefill phase, but also no significant additional overhead.**
> >
> > > **Q2**:  Evaluation on long generation scenarios
> >
> > More explanation about the experiments:
> >
> > 1. The average generated length was approximately **1500 tokens**.
> >
> > 2. Although there are some significant shortcomings in the above experiment, such as the limited sample size, we believe the results to som extent indicate that SKVQ has a good performance in long generation scenarios. In some cases, the diversity of the generated content due to quantization even results in higher scores compared to the full precision model.
> >
> > Should you have any further inquiries, please let us know and we are more than delighted to discuss with you and run more experiments for any pieces of your interests in our work.
> >
> > > **Q3**: Evaluation  on larger models?
> >
> > We evaluated SKVQ on Llama3-70B-Instruct on LongBench, which adopt GQA with $\text{NumHeads}=64, \text{NumKVHeads=8}$ (i.e. 8 queries share one KV-cache group). Our results show that for most tasks, SKVQ's accuracy loss is within 2%, and for some tasks, it is lossless. However, for a few tasks like `Qasper`, SKVQ experiences more significant accuracy loss. We believe this is largely because the full-precision model itself performs poorly on this task. We believe this experiment further demonstrates the generalization ability of SKVQ on models of larger size.
> >
> > ---
> >
> > We sincerely appreciate the time and efforts you have dedicated to reviewing our paper. Should you have any further inquiries, please let us know and we are more than delighted to discuss with you and run more experiments for any pieces of your interests in our work.

---

> > > ### Comment · Reviewer_QPwH · 2024-06-06
> > > **Thanks for the detailed response.**
> > >
> > > Thank you for the detailed response. My concerns have mostly been addressed.
> > >
> > > Regarding Q1, there are some real-world cases where long prompts are used with short generations, e.g. 4K/8K tokens in prefill and generating only 64 tokens. In these scenarios, the prefill can account for up to 20-30% of the end-to-end inference time, and this ratio could be even higher if decoding becomes faster, as suggested in the paper. While I believe that this paper is very useful in general cases, I wanted to note that there are still some corner cases that may require further investigation.

---

> > > > ### Author Response · Authors · 2024-06-07
> > > >
> > > > We are pleased that our response has addressed some of your concerns. We fully agree with the scenario you mentioned, especially in contexts like summarizing extremely long texts (e.g., 1 million tokens). In the prefill phase, due to the quadratic complexity, this part of the computation can indeed dominate the inference time. SKVQ may not provide significant acceleration for such scenarios. However, we do believe this limitation extends beyond the capability boundaries of most KV-cache quantization efforts.
> > > >
> > > > Overall, we greatly appreciate your detailed and insightful feedback.

---

### Decision · Program_Chairs · 2024-07-10

**Decision:**

Accept

**Comment:**

Quality
- Pro
  - Convincing results, well-motivated
  - Technically and experimentally rigorous and thorough
- Con
  - Too technical for NLP audiences

Clarity
- Con
  - minor grammar issues
  - explanations of key concepts and ideas could be clearer
  - notation and results not well explained

Originality
- Pro
  - novel method that augments other approaches

Significance
- Pro
  - Effective solution for a highly relevant practical problem
  - novel and likely good impact on the community and field

**Decision**

All reviewers agree that this work presents a novel, effective solution that is well-supported by the rigours and extensive experiments. The only concern that almost all reviewers have is with clarity. Some concepts, notation are unclear and there are other issues like grammar. Overall, this is a clear accept. I recommend that the authors to work on clarity of the draft for the paper-ready to increase the impact of this work.